# A Review on the Progress of Optoelectronic Devices Based on TiO_2_ Thin Films and Nanomaterials

**DOI:** 10.3390/nano13071141

**Published:** 2023-03-23

**Authors:** Shunhao Ge, Dandan Sang, Liangrui Zou, Yu Yao, Chuandong Zhou, Hailong Fu, Hongzhu Xi, Jianchao Fan, Lijian Meng, Cong Wang

**Affiliations:** 1Shandong Key Laboratory of Optical Communication Science and Technology, School of Physics Science and Information Technology, Liaocheng University, Liaocheng 252000, China; 2Zhejiang Province Key Laboratory of Quantum Technology and Device, Department of Physics, Zhejiang University, Hangzhou 310027, China; 3Anhui Huadong Photoelectric Technology Research Institute, Wuhu 241002, China; 4Shandong Liaocheng Laixin Powder Materials Science and Technology Co., Ltd., Liaocheng 252000, China; 5Instituto Superior de Engenharia do Porto, Polytechnic of Porto, Rua António Bernardino de Almeida, 4249-015 Porto, Portugal; 6College of Mathematics and Physics, Beijing University of Chemical Technology, Beijing 100029, China

**Keywords:** TiO_2_, optoelectronic devices, heterojunction, nanomaterial

## Abstract

Titanium dioxide (TiO_2_) is a kind of wide-bandgap semiconductor. Nano-TiO_2_ devices exhibit size-dependent and novel photoelectric performance due to their quantum limiting effect, high absorption coefficient, high surface-volume ratio, adjustable band gap, etc. Due to their excellent electronic performance, abundant presence, and high cost performance, they are widely used in various application fields such as memory, sensors, and photodiodes. This article provides an overview of the most recent developments in the application of nanostructured TiO_2_-based optoelectronic devices. Various complex devices are considered, such as sensors, photodetectors, light-emitting diodes (LEDs), storage applications, and field-effect transistors (FETs). This review of recent discoveries in TiO_2_-based optoelectronic devices, along with summary reviews and predictions, has important implications for the development of transitional metal oxides in optoelectronic applications for researchers.

## 1. Introduction

TiO_2_ nanomaterials are widely used in memory resistors, photocatalysis, sensing, and other fields because of their excellent electronic properties, simple preparation process, and high-cost performance. The three main TiO_2_ crystal forms are rutile, anatase, and titania. All three crystal structures consist of TiO_6_ octahedra but are connected in different ways. Of these ores, rutile and anatase have better stability and are more widely used. Between them, the anatase phase is the main phase with better catalytic performance, and the anatase phase TiO_2_ films show a smaller effective mass of electrons. Due to its thermal stability, wide band gap, and environmentally friendly properties, it is a promising material choice for energy conversion applications. Various forms of TiO_2_ nanostructures (e.g., transient nanorods (NRs), nanosheets, three dimensional (3D) nanostructures, and thin films (TFs)) have been used in various applications such as detectors, memories, high-efficiency hydrolysis, diodes, transistors, etc. Currently, nano-TiO_2_ is being deposited on a wide variety of base materials for the fabrication of optical and electronic products, for example, MoS_2_, NiO, diamonds, Al_2_O_3_, etc. Over the last few years, several reviews have been published on ultraviolet (UV) detectors and sensors. So far, there has been no thorough coverage of TiO_2_-based optoelectronic device applications. In this review, we explore the application of titanium dioxide nanoparticles in various fields, including sensors, field-effect transistors, memories, light-emitting diodes, and photodetectors. Based on the present advances, we propose issues and prospects for optoelectronic components made of TiO_2_-based materials that will guide the advancement of TiO_2_-based electro-optical components.

## 2. Optoelectrical Application Based on TiO_2_ Devices

### 2.1. UV Photodetector

The UV detector converts the UV signal into an electrical signal. It has the advantages of reduced background noise, high accuracy, and a simple structure without cooling, showing unique advantages in some fields. Therefore, UV detectors are used extensively for UV light guidance, guided missile signature, as well as shipboard communications, and have shown massive potential for strategic importance within the defense area [1]. One-dimensional nanomaterials have been widely studied in gas-sensitive materials, solar cells, photocatalysis, and especially UV detectors, because of their high aspect ratios, large area sizes, and good controllability [2]. The photo-catalytic effect of TiO_2_ is widely recognized, and much work is devoted to enhancing its photocatalytic efficiency by narrowing its band gap and improving its redox activity by delaying the compounding of photogenerated carriers to enhance its photocatalytic efficiency [3]. The wide-bandgap of TiO_2_ and its outstanding physics, chemistry, and light characteristics suggest that TiO_2_ films could be useful for photoconductive detectors of vacuum ultraviolet (VUV) radiation [4].

#### 2.1.1. UV Photodetectors Based on Nanoparticle Structures

Li et al. synthesized TiO_2_: graphitic diyne (GDY) nanocomposites by coating anatase phase TiO_2_ nanocrystals on GDY particles [5]. The speed was significantly improved compared with the MgZnO photodetectors (PD). GDY, a two-dimensional structure consisting of a benzene ring and a carbon–carbon triple bond, is a recent carbon isotope compound. Its combination with TiO_2_ suggested its potential application in various optoelectronic fields. Tunable integration of GDY nanoparticles (NPs) with various nanomaterials will create military and civil prospects. Zheng et al. proposed an ultrathin magnesium oxide layer as a hole barrier layer (HBL) between the titanium dioxide NP layer and the FTO conductive substrate, with the help of a magnesium oxide nanofilm, and the UV-PD based on TiO_2_ NPs had a high response rate and sensitivities [6]. After 36 h of hydrolysis at 0.2 M TiCl_4_, the MgO/TiO_2_ NPs UV PDs had a maximum sensitivity of 29.97 AW, and a high response time (rise time 0.02 **s**, decay time 0.06 **s**) and sensitivity of 10^4^. In addition, ultraviolet photodetectors have remarkable UV spectrum selection and excellent stability in the 365 nm range.

#### 2.1.2. UV Photodetectors Based on One-Dimensional Nanomaterials

Elsayed et al. developed a new porous aluminum oxide template (PAOT)/TiO_2_/TiN/Au photodetector of good reliability, low cost, and excellent performance by coating gold nanoparticles to the interface of the TiO_2_/TiN nanotube photodetector [7]. The response rate (R), detection rate (D), and external quantum efficiency (EQE) of the PAOT/TiO_2_/TiN photodetector at 390 nm were 453 mAW^−^^1^, 8.0 × 10 Jones, and 9.64%. The data indicated that the constructed PD has a very high level of photo response, which makes it promising as a broadband photodetector. Gao et al. constructed Ag/TiO_2_-Schottky diode thermoelectric photodetectors using thin, porous TiO_2_ with 9.8 × 10^10^ cmHz^1/2^/W high detection rate and 112 μs and 24 μs [8], the high sensitivity of rise and fall times. With microsecond response and high detection performance, porous Ag/TiO_2_ thermoelectric photodetectors have promising applications in various imaging, optical communication, and optical switching fields. Yang et al. prepared a novel all-solid heterostructure UV detector based on TiO_2_ nanoarrays and SiO_2_ dielectric layers [9] (Figure 1a). The addition of SiO_2_ had a significant influence on the response rate of the device. The response rate of the prepared TiO_2_/SiO_2_ hybrid devices to PD4 under UV irradiation was 1.15 A/W and 7.74 × 10^13^ Jones. Figure 1b,c shows the *I*–*V* curves of the TiO_2_/SiO_2_ hybrid devices. Through the analysis of the band map of detectors based on TiO_2_ nanoarrays (NAs) and TiO_2_/SiO_2_ hybridization, it can be verified that when the discontinuous SiO_2_ layer is inserted, a stable loss layer will be formed on the surface of TiO_2_ NAs, which reduces the defects on the surface of TiO_2_. Therefore, it is verified that the hybrid PD current described in the literature changes slightly. When a specific UV light is irradiated on the hybrid device, the photogenerated electrons and holes flow to the FTO electrode and the Ag electrode respectively. However, due to the existence of the loss layer, the electrons will be partially lost, so the loss rate of carriers increases. Compared with PD based on pure TiO_2_ NAs, the attenuation time is greatly shortened. (As shown in Figure 1d,e) and decreasing the responsiveness of the device. This device provides a solid foundation for the development of all-solid-state photodetectors.

The p-n-p monolayer graphene photodetector with titanium dioxide-doped nanotubes was proposed by Huang et al. [10]. Compared with a single layer of graphene, graphene integrated with titanium dioxide nanotubes increased the intensity of the photoreaction and generated photocurrent without a bias voltage. These devices are environmentally friendly and sustainable and are expected to be a promising choice of material for low-energy photovoltaic devices. Zhang et al. prepared TiO_2_ nanowires with a 100 nm average diameter of gold-doped nanoparticles using a simple solvothermal method [11]. The introduction of gold nanoparticles had a minimal effect on the energy band gap of pure TiO_2_, providing the sensor with unique physicochemical and electronic properties. In addition, the introduction of gold nanoparticles reduced the dark current, resulting in good sensing performance in the doped detector. The Au-TiO_2_ nanowires (NW) detector has the advantages of simple fabrication technology, low noise, and good comprehensive performance, which suggests the development of a UV image array [4]. Figure 2a shows the process for preparing and assembling a self-powered UV detector. Compared with TiO_2_ NRs/polyterthiophene (PTTh), the photocurrent of TiO_2_ NRs/Au/PTTh increased by a factor of 10, and the corresponding values of R and D increased from 77.39 μA/W and 2.155 × 10^9^ Jones to 1.894 mA/W and 1.666 × 10^10^ Jones (Figure 2b,c), respectively. This study offered a way to improve the operation of self-powered UV detectors, both technically and theoretically. The addition of gold nanoparticles changed the device’s energy level, leading to charged carriers. The band diagram of the TiO_2_ NRs/Au/PTTh heterojunction was analyzed, and the results suggested that Au NPs could enhance the performance of the photodetector. The use of photoelectric and thermoelectric effects can effectively increase the internal electric field, accelerate the charge migration, and reduce the compound rate, thus achieving high-performance UV photoelectric detection (Figure 2d,e).

#### 2.1.3. UV Photodetectors Based on Two-Dimensional Nanomaterials

Marilou et al. used pure titanium as a target material for VUV radiation photodetectors [12]. They prepared TiO_2_ films by reactive DC magnetron sputtering on the surface of soda lime glass (SLG) microscope slide substrates. TiO_2_ is a VUV photodetector, that can be post-processed by gamma irradiation to improve its light conductivity. TiO_2_ films have a wide forbidden bandwidth and good radiation resistance, opening up a wide range of applications for UV radiation detection. Gu et al. prepared TiO_2_ thin films by DC reactive magnetron sputtering and investigated the effect of target base sputtering distance on the properties of TiO_2_ thin films [1]. When the distance between target and substrate (D_TS_) was 80 mm, the TiO_2_ UV detector had a high responsiveness of 2.02 × 10^4^ A/W. The response speed was fast, the rising time (tr) was 0.98 s, and the falling time (tf) was 3.14 s. The method will guide the development of TiO_2_ UV detectors with high reaction speeds. Fang et al. successfully prepared TiO_2_ nanotubes by liquid-phase deposition using ZnO nanorod arrays as a template [2]. Then, they deposited Ag nanoparticles on the surface of TiO_2_ nanotubes and prepared UV detectors. The operation of the UV detector is shown in Figure 3e. After modification with Ag nanoparticles, the photocurrent density reached 91 µA/cm^2^ while the switching rate was 2251. At the same time, the response time was significantly shortened, with the rise and fall times reduced from 0.081 s and 0.05 s to 0.035 s and 0.025 s, respectively (Figure 3a–d). The UV-detector mechanism demonstrated that the Schottky barrier between Ag/TiO_2_ nanotubes can effectively promote the separation of photogenerated carriers and inhibit the photoelectron-hole complexation, thus improving the photogenerated carrier lifetime [2].

Zhang et al. fabricated NaTaO_3_/TiO_2_ heterojunctions by spin-coating NaTaO_3_ on TiO_2_ films and prepared UV detectors by depositing silver electrodes on the surface of the composite films [13]. The single TiO_2_ film detector had no photo response to UV light. The pure NaTaO_3_ sensor had a resistance-to-current ratio of only 3, significantly lower than the NaTaO_3_/TiO_2_ detector. Ezhilmaran et al. prepared and studied the all-oxide, highly transparent TiO_2_/MoO_3_ bilayer film for the first time on the platform of nano-anatase [14]. The film was composed using a simple solution method, and its application in ultraviolet detection was demonstrated. The bilayer-structured device exhibited good photo response and detection performance under very low-intensity UV illumination at 352 nm. TiO_2_ nanoparticles were used as the platform. The surface was two layers of heterogeneous junctions of α-MoO_3_, which have high transparency and the best light response characteristics under low bias voltage and weak illumination, so it is an ideal choice for spontaneous light detectors.

### 2.2. Light-Emitting Diodes (LEDs)

Due to its low cost, light weight, thinness, and wide viewing angle, LED has attracted a lot of attention. LED is a new electronic light source, with common energy usage, long life, small volume, fast on/off speed, etc. [15]. LEDs are not merely used as an indicator of low energy consumption, but also as an alternative to conventional indoor/outdoor light sources. It is considered the next primary light source substitute for the current lighting regime. The demand remains high for more energy-efficient LEDs to extend battery life and penetrate more markets [16]. Nanostructured TiO_2_ is chemically and thermally stable, less sensitive to moisture and oxygen, and has good optical properties. Compared to organic materials, TiO_2_ has a higher electron mobility, which results in efficient electron transport and increases the chance of charge complexation [17]. Therefore, TiO_2_, as a promising material for electron transfer, has a variety of applications, especially in luminescence. The optical and optoelectronic properties of poly [2-methoxy,5-(2-ethylhexoxy)-1,4-phenylenevinylene] (MEH-PPV) composite films were investigated by Asbahi et al. [18]. The research indicated that the luminous intensity of MEH-PPV was improved by the improvement in charge transfer efficiency. The results proved that the luminescence and attenuation lives of the MEH-PPV/SiO_2_/TiO_2_ nanocomposites (STNCs) film were smaller than those of the original MEH-PPV film. The attenuation period was lower, which further proved that the MEH-PPV has high transmission efficiency and does not produce static explosions. Lee et al. studied two kinds of SiO_2_/TiO_2_ distributed bragg reflectors (DBRs) with ITO ohmic contact in InGaN/GaN flip-chip LEDs (FC-LED) [19]. The structure of a DBR enhanced the reflectance by more than 95% at different incident angles. At 200 mA, the output power of DBR 2 was 135 mW, while the FC-LED with DBR 1 was 114 mW. The results demonstrated that the EQE values of the Lee’s FC-LED and DBR 1 were consistent. Su et al. simulated the emission characteristics of perovskite light-emitting diode (PeLED), simulated its internal aluminum core radius, dielectric layer thickness, external silver shell thickness, dielectric layer material, and luminous layer spacing, and analyzed its brilliant characteristics [20]. The results demonstrated that when the shell thickness of silver increases, the ${|\omega }_{-}^{-}$〉 type will appear to have a red shift, and when the dielectric layer thickness increases, the ${|\omega }_{+}^{-}$〉 type will appear to have a slightly blue shift. The simulation results indicated that the use of AL-TiO_2_-Ag bimetallic particles can improve the luminescence characteristics of LEDs. Bimetallic nanoparticles have great potential in the photoelectric field. Song et al. used silane (C_3_–C_16_) with different carbon chain lengths to control the surface properties of TiO_2_ and evaluated its optical properties by changing the dispersion degree of TiO_2_ particles [21]. The dispersion of TiO_2_ modified by silane was 1.6–2.5 times that of original TiO_2_. With the increased carbon chain length (C_3_–C_6_) of silane, the dispersibility of TiO_2_ also increased. On the premise of not reducing the radiation flux, adding TiO_2_ into the filler can obtain a higher luminous flux and a lower correlation color temperature (CCT) value. In addition, the higher the dispersion of TiO_2_, the better its optical properties and color uniformity. Carbon nanotubes (CNTs) /TiO_2_/SiO_2_/p-Si metal oxide semiconductor (MOS) structures for LEDs were synthesized by Ashery et al. [22]. This study allowed one to control and adjust the value of capacitance and its signal polarity to positive or negative values, enabling tunneling behavior and *I-V* measurements of negative resistance. The capacitance varied with temperature, voltage, and frequency, with a maximum weight of 2.12 × 10^−9^~2.4 × 10^−7^ F, and in the low-frequency range of 10^3^–10 Hz with a maximum weight of 4 × 10^5^~5 × 10^−3^ F. Kim et al. reported a new study of the Al-doped TiO_2_ interface layer for improving charge balance and quantum-dot light-emitting diodes (QLED) luminous efficiency [23]. The photoluminescence in the Al-doped TiO_2_ (ATO) film was analyzed, and the oxygen vacancies and defects in the ATO film were determined. The optimized ATO layer of QLED improved device performance with a luminous efficiency of up to 119,516 cd m^−2^. In conventional devices (QLED with pure TiO_2_ and QLED with the ATO layer), it is 3.3, 7.6, and 12.0 h, respectively. The characteristic curves of the transistor and ATO interface layers were analyzed by UPS and PL deconvolution spectra. The results indicated that the holes injected by the ITO anode easily crossed the intermediate band gap of the ATO layer. The results demonstrated that QLED with ATO has a good charge balance because of the large number of hole injections, which can prolong the service life of the device. In this paper, it was found that inserting the ATO interface layer between the QLED indium tin oxide (ITO) anode and V_2_O_5_ hole injection layer (HIL) can realize efficient hole injection. Kim et al. reported an efficient QLED that uses lithium-doped TiO_2_ NP instead of an electron transport layer (ETL) [24]. Electronic structure analysis of original and lithium-doped TiO_2_ NP showed an increase in the conduction band minimum (CBM) energy level because of lithium doping, which increased the barrier to the energy injection of electrons into the ETL from the electrode. When the ETL was TiO_2_ NP doped with 5% Li, the green QLED had a maximum brightness of 169,790 cd m^−2^ and an EQE of 10.27%, which was the highest electroluminescence (EL) value for a non-ZnO inorganic ETL QLED. Compared with green quantum dots (QDs), the CBM level of TiO_2_ enabled electrons to smoothly pass through the emitting layer (EML) of ITO. In contrast, its maximum depth valence band maximum (VBM) energy level (−7.95 eV) effectively prevented the holes in the interface between the EML and ETL. These results suggested that Li-doped TiO_2_ NP has a good prospect in QLED as a solution for inorganic ETL.

### 2.3. Sensing Applications

Various materials based on metal nanoparticles [25] have been used to construct various sensors [26] and represent a promising way to improve sensor selectivity and sensitivity. Among the different nanoparticles, metal oxides can promote redox reactions with the measured substances and change their electrical conductivity, thus becoming a hot research topic in recent years. In addition, because of the interaction between the device and the target molecule, electron transfer can be increased, improving the sensor’s performance [27]. TiO_2_ is widely used in sensing applications because of its excellent chemical properties, large specific surface area, catalytic efficiency, electrical conductivity, high physical strength, and other properties.

#### 2.3.1. Biosensor

Ognjanović et al. proposed a new technology that immobilized the enzyme on the electrode to improve its structure, establishing the impedance type glucose biosensor [26]. Nanoparticles with carboxyl graphene were prepared by silanizing titanium dioxide nanoparticles with (3-aminopropyl) triethoxysilane (APTES). The combination effectively improved the electrode surface and provided a good foundation for the establishment of a point-of- care device [27]. TiO_2_ nanoparticles with Zn-doped levels of 2.5, 5, and 10 mol% were prepared and characterized by Meesombad et al. [28]. using the solution combustion method to explore their promising applications in sensors. A lower peak current degradation was found during more than 200 tests. Thus, 2.5 mol% Zn-doped TiO_2_ nanoparticles have the potential for use in the detection of glutamate. A biosensor based on Cu_2_O-TiO_2_ hybrid nanostructures was investigated by Khaliq et al. [29]. The sensitivity of TiO_2_ nanotubes modified by Cu_2_O nanoparticles (6034.04 μA mM^−1^cm^−2^) was five times higher than that of unmodified TiO_2_ nanotubes. This paper demonstrated that Cu_2_O NP-modified TiO_2_ nanotubes have good stability, reproducibility, and selectivity. To improve the Raman intensity of methylene blue (MB) and establish a sensitive uracil DNA glycolase surface enhanced Raman scattering (UDGSERS) biosensor, Huang et al. used the catalytic hairpin assembly (CHA) method to synthesize Ag/TiO_2_ nanocomposites for charge transfer (CT) and surface plasmon resonance (SPR) effects [30]. Using Pb^2+^ to convert trace UDG into large amounts of trigger DNA can further improve the sensitivity of SERS detection. Therefore, this experimental design will guide the preparation of SERS composite substrates for biomolecular detection and clinical analysis. Xu et al. reported a biosensor for detecting TiO_2_-CTF-enhanced gold nano-islands (TiO_2_-CTFe-AuNIs) from exosomes of glioma cells (GMs) [31]. The combination of TiO_2_ cylindrical films and gold nano-islands significantly improved their plasmonic sensing capability in the visible region. The limit of detection (LOD) of GM derivatives was 4.24 × 10^−3^ μg mL^−1^. Compared with the SAM-Aunis localized surface plasmon resonance (LSPR) biosensor, the sensor had a 2.5-fold higher sensitivity. The existing TiO_2_-CTFe-Aunis plasma biosensor can sensitively detect big3 in the exosomes derived from GMs to monitor the effect of maternal GMs on anti-tumor drugs. This has a significant impact on the recovery from glioma after treatment. Jeong et al. proposed a non-enzymatic glucose sensor in which conducting polymer films were prepared by synthesizing high-purity titanium dioxide nanoparticles from isothermal ionization, directly depositing them on the substrate, and then by electrochemical deposition of chitosan–polypyrrole (CS-PPY) [32]. The sensitivity of the CS-PPY/TiO_2_ biosensor was 302.0 μA mM^−1^cm^−2^ (R^2^ = 0.9957) with a detection limit of 6.7 μM. Experimental data proved that the device provided a strong basis for the future development of glucose detectors. Yoo et al. fabricated DNA biosensors using highly purified titanium dioxide nanoparticles deposited on interdigitated electrodes (IDE) by hot plasma [27]. The detection of the microflow on the surface of the biosensor using a picometer enabled qualitative identification of DNA with high sensitivity, specificity, and characterization. The advantage of this biosensor is that it is simpler and less noisy than traditional electrochemical methods and does not require chemical components other than DI water. This research will likely be used to develop a diagnostic capability for high-risk infectious diseases such as foot-and-mouth disease in African pigs without complex field pre-treatment and cloning processes. Photoelectrochemical (PEC) photodetectors have multiple functions, such as light capture, charge carriers, and surface chemical reactions. Therefore, it is essential to choose suitable photoactive materials for constructing efficient PEC biosensors. Xu et al. constructed a TiO_2_/polydopamine (PDA) core/shell nanorod array-based fast interfacial transfer photo pole as a stable and highly sensitive PEC glucose biosensor [33]. Compared with the pure TiO_2_-based PEC biosensor, this PEC biosensor had better optoelectronic performance and better glucose detection. The sensitivity of glucose detection was 57.72 mA mM^−1^cm^−2^, the linear range was 0.2–1 mM, the detection limit was 0.0285 mM (S/N = 3), the dynamic range was 1–6 mM, and the sensitivity was 8.75 mA mM^−1^cm^−2^. PDA is important for the rapid transfer of photogenerated carriers between the interface between TiO_2_ and the enzyme. Lu et al. prepared a biosensing platform by moving TiO_2_/Ti_3_C_2_T_x_ (TiO_2_/TiCT) nanocomposite-covalent organic polymers (NUF) heterostructures onto carbon cloth electrodes (CCEs), where TiO_2_/TiCT and NUF-derived polar functional groups could enhance the interaction with the CCE surface [34]. This made the surface of the material and the CCE more solid. This new biosensor can prevent the adsorption of non-specific proteins on the electrode. Therefore, TiO_2_/TiCT/NUF can be used as a bioactive molecular sensor for medical diagnosis and food surveillance [34]. Yang et al. used PDA and aminofunctionalized graphene quantum dots (N-GQDs)/glucose oxidase (GO_x_) to modify TiO_2_ nanotubes (NTs) and established a highly sensitive photoelectric double receptor biosensor [35]. Based on the synergistic effect of organic and inorganic semiconductors, the double electron acceptor series heterojunction biosensor is a biosensor testing platform with broad application prospects. As a result of the complex changes in blood glucose concentration in diabetic patients, long-term monitoring is required. Therefore, the monitoring range of the sensor should be wide and stable. Wang et al. prepared a TiO_2_/SrTiO_3_/PDA/GO_x_-PEC biosensor by a simple hydrothermal method using GO_x_ as a model and PDA as a binder [36] (Figure 4a). The sensor has a linear range of 0–32 mM and a sensitivity of 5.37 mA mM^−1^cm^−2^ (Figure 4b). By studying the sensing mechanism of this biosensor, it was found that when glucose is acted on by GO_x_ and oxygen, it will be decomposed into gluconic acid and H_2_O_2_, and the H_2_O_2_ will be oxidized through the holes in the electrode, thus forming a photocurrent (Figure 4c). Using TiO_2_/SrTiO_3_/PDA/GO_x_ as a new biosensor, the relative standard deviation in the range of 0-20 mm was only 1.22%; this will provide a reference for the future development of the blood glucose detection sensor system.

#### 2.3.2. Gas Sensor

Noh et al. developed a NO_2_ gas sensor operating at room temperature using photoactivation and charge separation of TiO_2_ NRs and Pt NPs [37]. TiO_2_ NRs were prepared by the well-known hydrothermal preparation method, which has many advantages over other preparation methods. The sensing performance of the TiO_2_ NR sensor after 2 h annealing was much better than that of the TiO_2_ NR sensor after 1 h or without annealing. It can be concluded that increasing crystallinity through heat treatment is an effective way to make photoactivated electrons reach the surface. The health management of carbon-containing combustion equipment can be achieved by monitoring the CO concentration in the exhaust gas. Chen et al. prepared porous TiO_2_/CeO_2_ detection gas sensors with the hydrothermal flame annealing method with different TiO_2_ coupling amounts using band gap engineering [38]. The combination of the hydrothermal method and high-temperature annealing simplifies the material coupling process. Cheng et al. developed a new NH_3_-type novel sensor based on TiO_2_/WO_3_·H_2_O (TW)-type heterojunctions [39]. The invention proposed a heterojunction structure composed of titanium dioxide and WO_3_·H_2_O, which provided a rich theoretical basis for developing this type of sensor. Li et al. prepared WO_3_-TiO_2_ composites as hydrogen-sensing elements and applied the heterojunction effect generated by combining WO_3_ and TiO_2_ to hydrogen sensing at room temperature [40]. The sensor material designed in this experiment had good stability, repeatability, and selectivity. These experimental results indicated that the WO_3_-TiO_2_ heterojunction with a 4.0 wt% mesoporous structure has excellent potential as a sensitive element. Cao et al. successfully synthesized three titanium dioxide samples with different crystalline phases using a simple hydrothermal method and prepared acetone sensors to study their gas-sensitive properties [41] (Figure 5a). When the three TiO_2_ nanomaterials were evaluated as gas sensors, rutile TiO_2_ nanorods exhibited the highest acetone sensing performance compared to anatase TiO_2_ nanoparticles and rutile TiO_2_ nanorods (Figure 5b). The rutile TiO_2_ nanorods (*Ra/Rg* = 12.3) responded to 100 PPM acetone 3.0 times more than the rutile TiO_2_ nanorods and 5.3 times more than the brookite TiO_2_ nanorods. The rutile TiO_2_ nanorod sensor showed high sensitivity (12.3), a short response time (3 s), good stabilization, and good selectivity for different volatile organic compounds (VOCs).

Wang et al. prepared PD-doped CoTiO_3_/TiO_2_ (PD-CTT) nanocomposites by hydrothermal synthesis and calcination of a TiO_2_ precursor using the TiO_2_ precursor as raw material [42] (Figure 6a). The PD-CTT-based sensor showed good response performance for benzene gas by testing different gases at 10 ppm at room temperature; the excellent selectivity of the PD-CTT-1 composite sensor for benzene is shown in Figure 6b. When benzene reacts with O_2_, a large number of electrons will be produced, which will destroy the balance of the Fermi energy level. The electron transfer between CoTiO_3_ and TiO_2_ can facilitate the reaction of benzene molecules because of the presence of heterojunctions, thus contributing to the sensor’s sensitivity to benzene gas (Figure 6c). This study will open new avenues for developing highly accurate and reliable benzene gas sensors that show great potential for applications in benzene hazard protection.

#### 2.3.3. Electrochemical Sensor

Electrochemical analysis has benefits such as low instrument cost, simple sample preparation, small size, fast reaction speed, high sensitivity, and good selectivity. Precise identification of phosphorylation sites and their quantification are essential to understanding pathogenic mechanisms and predisposing factors. Chen et al. developed an electrochemical sensor capable of rapid and sensitive detection of phosphopeptides using the specific binding of titanium dioxide nanoparticles to phosphopeptides [43]. The electrochemical sensor is a good application platform for detecting phosphopeptides in clinical settings. The preparation of enzyme-induced glucose sensors based on mesoporous titanium dioxide nanoparticles (MTNPs) was investigated by Amri et al. [44]. The fourier transform infrared (FTIR) spectra showed corresponding characteristic peaks, indicating the successful immobilization of GO_x_ on the surface of MTNPs-2. The glass carbon electrode (GCE)/MTNPs-2/GO_x_ electrode has a suitable direct electron transfer (DET) pathway between the GO_x_ and the modified electrode and can be used for glucose detection. The sensor has a sensitivity of 0.4098 μA mM^−1^cm^−2^, a linear range of 0.1 to 1 mM, and a relatively low detection limit of 73 μM. Kusior et al. successfully prepared an electrochemical detector with a TiO_2_-Fe_2_O_3_ coupling system as an electrochemical detector for reinforcing materials [45]. They measured the electrocatalytic reaction of the prepared electrode to hydrazine in 0.1 M NaOH. The increase in peak anode current with increasing hydrazine concentration indicated that the oxidation process of hydrazine was irreversible. The proposed TiO_2_-Fe_2_O_3_ coupling system provided a suitable material for a stable, sensitive, and easy-to-use hydrazine sensor. To better detect the content of temptation red in food, Li et al. developed a new electrochemical sensor based on TiO_2_/electro-reduced graphene oxide nanohybrids (TiO_2_/ErGO) nanocomposites [46]. The detection limit was estimated to be 0.05 m when the signal-to-noise ratio was 3. The TiO_2_/ErGO/GCE nanocomposite has become a popular candidate material for food colorant detection because of its high sensitivity, anti-interference performance, repeatability, reproducibility, and stability. The photoelectrochemical detection platform based on nucleic acid aptamer-modified sandwiched ZnCuInSe/Au/TiO_2_ nanowires was developed by Geng et al. [47]. Due to the high photocurrent density, the sensor has a high sensitivity with a detection limit of 0.2 nM.

### 2.4. Transistors

High-performance thin-film transistors (TFTs) have become a significant component of electronic devices. High dielectric materials facilitate the realization of low-voltage and high-performance thin-film transistors because of their large area capacitance and sender-type electron traps [48]. Compared to silicon-based thin-film transistors, titanium oxide has high transparency [49] in the visible range, high chemical stability [50], non-toxicity [51], and earth abundance [52]. It can be used as an alternative channel material for metal oxide thin-film transistor applications. TiO_2_ is expected to become a vital channel material for TFTs because of its excellent chemical and mechanical stability, non-toxicity, and abundance of resources.

#### 2.4.1. Junction-Type Field-Effect Transistors

Electrochemical synaptic transistor (SynT)-based artificial synapses are of great interest in massively parallel computing. Yet, most SynTs are still plagued by degradation limitations and high energy consumption. To solve the above problem, Nguyen et al. proposed a low-power all-solid synthetic device fabricated using wafer-level microfabrication technology [53]. Electrochemical impedance spectroscopy (EIS) and rate capability tests further confirmed the resilience of TiO_2_ membranes at different scan rates; these results emphasized the great potential of Li_x_TiO_2_-based SynT for efficient neuromorphic applications. Gakhar et al. prepared FETs based on p-TiO_2_ nanoparticles/graphene oxide heterostructures for detecting VOCs [54] (Figure 7a). The p-anatase-type TiO_2_ nanoparticles prepared by the sol–gel method were injected into a small number of layered graphene/GO channels as the channel material. Figure 7b illustrates that the sensor showed stable characteristics at higher gate-source voltages. Figure 7c shows that because of the change in environment from air to VOC, when *V_GS_
*= 0, the Fermi level shift of the GO channel was estimated as Δ*E*. When *V_GS_
*> 0, extra electrons [n(*V_GS_*)] passed through the field-effect injection channel, and the Fermi level in GO moved upward, thus reducing the effective hole concentration and leakage current in the air environment. The field-assisted sensitivity amplification technology on the p-TiO_2_ NPs/rGO FET sensor improved detection of the lower limit of VOCs.

Qin et al. made high-electron-mobility transistors (HEMTs) with and without the TiO_2_ layer and set the TiO_2_ layer as the protective layer between the AlGaN barrier layer and the SiN passivation layer [55]. From the perspective of Vth bias, HEMT with TiO_2_ had a greater saturation tendency than HEMT without TiO_2_ at high voltage *V_Gs_*. The TiO_2_ embedded layer can reduce the damage caused by etching, which provides a valuable reference for the popularization of AlGaN/GaN HEMT. Qi et al. prepared TiO_2_ nanowires using a hydrothermal method, which showed n-type semiconductors because of inherent oxygen vacancies [56]. TiO_2_ nanowire crystals reasonably simulated the plasticity of synapses, such as paired-pulse distribution and high-pass filtering. Therefore, it was demonstrated that TiO_2_ nanowire transistors have promising applications in future neural networks. Zhang et al. proposed to design p-i-n junction synaptic transistors (JST) (p-i-n JSTs), with a composite structure of various polymer materials [57] (Figure 8a). The recognition accuracy of the extended MNIST (EMNIST) pattern by p-i-n JST was ≈ 93%, and there was no noticeable change after 6 months (Figure 8b). The P3HT/TiO_2_ junction was banded with a bias voltage (Figure 8c). When *V_D_* and *V_G_* are positive, most carriers can form a current on the contact surface because of the influence of the diffusion mechanism. The positive pulse is injected into P3HT/PEO film to form the reverse layer, which improves the carrying capacity of TiO_2_ film and the long-term plasticity of TiO_2_ film. This device can control the back-and-forth flexure driven by polymer materials to adapt to the input of various synapses.

#### 2.4.2. Insulated Gate Field Effect Transistors

TiO_2_ has the advantage of being relatively inexpensive compared to other oxide dielectrics and is an excellent candidate for high-k applications. Yang et al. successfully modified TiO_2_ with polyvinylpyrrolidone (PVP) using the inorganic–organic composite media method to obtain TiO_2_:PVP mixed media [48]. The roughness, thickness, and band gap increased as the PVP concentration increased, the leakage current density decreased and then increased, and the area capacitance and relative dielectric constant (*k*) decreased. Composite TiO_2_:PVP films with these properties have enormous promise for wearables, synaptic bionics, and intelligent computing. Subramanian et al. explored ion-gated transistors (IGTs) based on TiO_2_ active layers that were prepared by thermal evaporation deposition for dense films and by solution casting for porous films [58]. The thick film-based transistors showed about two orders of magnitude higher electron mobility than the porous film-based transistors. For the solution-treated films, the presence of Li^+^ in the selected-pass medium resulted in a significant increase in electron mobility. Zhang et al. systematically investigated the effect of zirconia gate dielectric thickness on the electrical properties of TiO_2_ thin-film transistors [59] (Figure 9a,b). Figure 9c shows the equivalent parallel conductance (*Gm*/*ω*) of the samples measured at different frequencies versus the applied voltage. As a result of the electronic structure of TiO_2_ and ZrO_2_ themselves, a clear asymmetric energy band shift was observed in energy band diagrams (Figure 9d). By analyzing the thickness of ZrO_2_, the influences of leakage current, oxide capacitance, oxide charge and interface trap were found, which indicated that the ZrO_2_ dielectric layer has good flexibility, can be applied to other channel materials, and can be used as a reference for evaluating the material quality of different dielectric materials.

Yokoyama et al. investigated the effect of V_M_ on the synthesis of a-TiO_2_ and a-Al_0.74_Ti_0.26_O_3_ using atmospheric pressure thin-mist chemical vapor deposition (CVD) [60]. The parameters of the FETs were enhanced using reticular biased a-TiO_2_ and a-Al_0.74_Ti_0.26_O_3_ films as the gate dielectric layer and mechanically exfoliated bulk single-crystal MoSe_2_. These results indicated that the applied grid bias effectively obtained high-density networks and better junction properties at the C–Si interface when a-TiO_2_ and a-Al_0.74_Ti_0.26_O_3_ thin films were synthesized by the atomic phase CVD method. Jung et al. proposed a high electron mobility transistor using a TiO_2_/SiN double-gate insulator [61]. Compared to the device with a SiN gate insulator, the device with dual-gate adiabatic showed improved gate controllability and channel electron density because of the high dielectric constant of TiO_2_. The device with the dual-gate insulator showed a higher breakdown voltage because of better electric field dispersion. In conclusion, devices with TiO_2_/SiN double-gate insulators are the best candidates for the usual turn-off of AlGaN/GaN MIS-HEMT. Solution-treated, top-contact, bottom-gate tin dioxide TFTs were prepared using the TiO_2_/Li-Al_2_O_3_ double-stacked gate dielectric of Pal et al. [62]. They showed an operating voltage within 2.0 V. The results indicated that the higher surface capacitance and lower leakage current density of the double-layer dielectric material significantly improved the overall performance of the thin-film transistors. This research opens new paths for developing TFT device performance using appropriate gate dielectric layering techniques. Using a low-temperature processing technique, Zhang et al. successfully fabricated a one-voltage TiO_2_ thin-film transistor [63]. This one-volt TiO_2_ thin-film transistor provides a broad scope for portable and flexible electronic products. In addition, this research revealed the possible carrier transfer mechanism in TiO_2_ thin films, which leads to a better understanding of the properties of TiO_2_ materials and thus provides a reference for the future development of TiO_2_-based electronic and optoelectronic devices. Enhanced/depletion mode (E-/D-mode) TiO_2_ TFTs were proposed by Zhang et al., illustrating the fabrication process of E/D mode TiO_2_ thin-film transistors [64]. The O_2_-annealed TiO_2_ TFTs exhibited higher performance with higher mobility (μ), higher *I_ON_/I_OFF_*, and lower SS than the N_2_-annealed TiO_2_ TFTs. This was due to the passivation produced by O_2_ annealing, which reduced the oxygen vacancies at the channel–oxide interface, thereby increasing the migration rate and reducing subthreshold swing (SS). Controlled O_2_/N_2_ annealing techniques are available as an alternative method to achieve E/D mode TFT, which provides greater flexibility in applications such as the preparation of field-effect tubes.

### 2.5. Memory Applications

As a result of its excellent endurance, fast storage speed, and ultra-high storage density, resistive random-access memory (ReRAM) has received much attention [65]. Resistor-switched random-access memory offers several advantages over traditional complementary metal oxide semiconductor (C-MOS)-based non-volatile memories, such as fast read/write speeds, high density, and low operating voltage, which make it a hopeful choice for future generations of non-volatile memories [66]. Metal oxides play a significant role in many aspects of physics and materials science. The transport of oxygen vacancies is a major factor in the conduction mechanism of memory elements. Transparent memory offloading (TMO)-based resistive switching non-volatile memory offers excellent storage performance, and many materials have already been tested for ReRAM applications. The binary metal oxides are very attractive among these resistive switching materials, in particular, TiO_2_, which is the most common oxide in ReRAM. It is essentially an insulator, which means very high resistance [67]. In addition, TiO_2_ is compatible with C-MOS and is an inexpensive, non-toxic, and chemically stable material. In particular, because of their excellent chemical and physical properties, one-dimensional TiO_2_ nanomaterials, which correspond to nanotubes and nanowires, show great promise for optoelectronic and nanoelectronic applications [65]. For the high-throughput study, TiO_2_ was selected as the benchmark material to establish a credible screening of ReRAM materials.

#### 2.5.1. Thin-Film Structured Memory

More et al. developed a photo-resistive switching memory consisting of a Pt/TiO_2_/Al junction structure by exploiting the optical sensitivity of nano-TiO_2_ [68]. Using the inherent photoconductive properties of TiO_2_, highly stable and reversible unipolar resistive switching with a long retention time (~10^4^ min) and stable duration (>300 cycles) was observed in nanocrystalline TiO_2_ films with a R*_ON_*/R*_OFF_* ratio of ~10^2^ at a read voltage of 0.5 V. The results suggested that reducing the clearing voltage of TiO_2_ films induced by optical illumination provides a new avenue for designing multifunctional photoactive nonvolatile device technologies. TiO_2_:Co bilayers and TiO_2_:Co films were prepared using DC magnetron co-sputtering at room temperature and 473 K by Quiroz et al. [69]. The experimental results demonstrated that the different resistive states generated by the applied field lead to lower energy consumption and higher storage capacity. Therefore, alternative configurations based on TiO_2_:Co bilayers and TiO_2_:Co films may be a nanomaterial with a wide field of application for the magneto-control properties of nonvolatile memories (NVM) resistive switching. Neuromorphic ones have advantages that differ from traditional computing architectures, such as the integration of data processing between the central processing unit (CPU) and memory. Since its underlying computing system mimics the information processing capabilities of the human brain, it is capable of performing many data operations simultaneously. Long-term and short-term memory are mainly controlled by the weight of synapses, which are key in brain memory. Yang et al. prepared and characterized Ti/TiO_2_/Si devices with different doping concentrations [70]. The Ti/TiO_2_/p^++^Si device with long-term memory was an interface-type bipolar resistive switch. The enhancement and weakening of different amplitude pulses were demonstrated with progressive resistive switching. In addition, when conductance was used as a network weight, 85% recognition accuracy was obtained in the neuromorphic system simulation. The pulse amplitude controls the dynamic range of the pulse. In the early 1960s, the phenomenon of resistance switching (RS) was discovered in various materials. The renewed interest in bipolar resistance (BRS) in SrZrO_3_ was reported in 2000. Over the past two decades, various materials have shown viable RS properties, which could make them potential candidates for the next generation of high-density nonvolatile storage devices. Yan et al. inserted a TiO_2_ layer between the bottom electrode of Al and the TiO_1.7_ layer to contact the top electrode of the Al nanometer-thick stoichiometric TiO_2_ layer [71]. Although the structural changes were small, the improvement in electrical properties was significant. With a switching state resistance ratio of 20, the resistance value can be maintained for up to 30,000 DC scan cycles and 10^6^ AC pulse switching cycles. In addition, the device is electroformed-free and displays full area-based switching characteristics. The device exhibits nearly linear enhancement and suppression characteristics under repeated pulse voltages, remarkably improving a neural network consisting of Al/TiO_1.7_/TiO_2_/Al storage cells in terms of accuracy. Memory structures have an important role to play in providing powerful processing power to integrated circuits. The synthesis of TiO_2_ thin films on Pt/TiO_2_/SiO_2_/Si (SSTOP) substrates by physical vapor deposition was first reported by Alsaiari et al. [67]. These techniques facilitate a variety of material library designs and rapid synthesis and permit library screening in a fast sequential or parallel fashion. CVD-based resistive memory has been successfully applied to this combined approach. A promising non-volatile memory device, the nano-resistive random-access memory (RRAM), has attracted much attention recently. Hu et al. used the sol–gel rotation and coating technology to prepare a resistive random-access memory based on TiO_2_ films [72]. By applying continuous positive (0–2 V) and negative (0–1.6 V) voltage scanning to study the synaptic performance, it was concluded that the devices manufactured exhibited “learning experience” behavior. The synaptic function of TiN/Ti/TiO_2_/SiO_x_/Si resistive memory was studied by Cho et al. [73]. The highly nonlinear *I*–*V* profile in the low-resistance state (LRS) generated by the SiO_x_ tunneling potential promoted a high-density synaptic array. ReRAM has been increasingly investigated as a new generation of memory with the advantages of small cell size and fast read/write speeds. Heo et al.’s amorphous InGaZnO (a-IGZO) oxide semiconductor was used as an active layer to prepare memory with a heterojunction structure [74]. It was found that the cumulative capacitance value of the ReRAM increased the most in a non-volatile manner after annealing at 400 °C. Thus, the results of this study indicated that the best electrical properties in the ReRAM at 400 °C were produced when the a-IGZO oxide active layer was post-treated with the optimal composition of the a-IGZO film.

#### 2.5.2. Non-Film Structured Memory

A flexible resistive switching memory based on TiO_2_ nanorod arrays was developed by Xue et al. with little variation in the operating voltage and switching ratio electrical properties of the device over a bending radius range of 12~5 mm [66]. The excellent performance of these devices demonstrated that the TiO_2_ nanorod array resistive switching device opens the possibility of achieving high-performance flexible memory. Yu et al. directly synthesized single-crystal rutile TiO_2_ nanowires by a simple low-temperature hydrothermal method [65]. Self-rectifying and shapeless non-volatile memory behavior derived from non-volatile memory was demonstrated for the TiO_2_ nanowire memory devices prepared in this study. This work indicated that titanium dioxide nanowire memory devices will be a promising candidate for the next generation of ultra-low power non-volatile memories. Sun et al. designed a graphene transistor-based photovoltaic memory [75] (Figure 10c), which was decorated with titanium dioxide nanoparticles and prepared by a low-temperature, easy-to-solve method using UV light programmable non-volatile optoelectronic memory (Figure 10a,b). Since the N-dopant-induced N-TiO_2_ nanoparticles (N-TiO_2_ NC) have a solid hole-capture ability, it was possible to record non-volatile information at multiple energy levels by precisely controlling the incident light dose. Thus, positive gate voltage eliminated the programmed state by promoting the compounding of stored cavities in N-TiO_2_ nanotubes (Figure 10d). This research demonstrated that trap engineering is important in information storage and provides an alternative route to non-volatile optoelectronic storage.

Kumari et al. prepared composite devices by hydrothermally growing titanium dioxide nanosheet (TiO_2_-NS) on FTO substrates and spin-coating polymers on FTO substrates [76]. The modified nanosheets exhibited attractive properties, such as worm/RRAM and modified diodes. This work showed how the new properties of the nanosheets can be used to restore the material’s properties by placing the polymer on the grown nanosheets. Among various one-dimensional nanostructures, different morphologies of titanium dioxide nanostructures have attracted much attention. To investigate the synergistic effect between TiO_2_ anatase and rutile mixed-phase interfaces, Bamola et al. reported mixed-phase TiO_2_ nanostructures (MxPh-TNs) and their growth through mixtures of grafted metal nanoparticles (MNPs) [77]. Figure 11a shows a schematic representation of the formation of this nanostructure. The increase in rutile content in platinum (Pt)/mixed-phase TiO_2_ hybrids (MX-TNHs) and Pd-Pt/MX-TNHs led to improved current ions (2.14 × 10^−5^ A) and *I_OFF_* (7.13 × 10^−5^ A), as well as current ions (2.13 × 10^−3^ A) and *I_OFF_* (6.77 × 10^−4^ A) of the devices at 3 V (Figure 11b). This study proposed a unique approach to improving device performance by grafting MNPs on MxPh-TNs to improve interfacial properties.

Pandey et al. prepared Ag NP-coated TiO_2_ NW array structures by the swept-angle deposition technique for capacitive storage applications [78]. Compared with bare TiO_2_ NW, the Ag NP-covered TiO_2_ NW had enhanced photoluminescence because of the high carrier presence. The Ag NP-covered TiO_2_ NW showed a significant enhancement in *C-V* hysteresis compared to the bare TiO_2_ NW NVM device because of carrier capture. The excellent durability and retention properties make this device structure a suitable candidate for capacitive memory applications. The resistive conversion properties of bismuth-doped anatase type TiO_2_ nanostructures prepared on Nb:SrTiO_3_ (Nb:STO) substrates were reported by Bogle et al. [79]. Figure 12a shows the synthesis process of TiO_2_ nanostructures. These nanostructures were prepared from Bi_4_Ti_3_O_12_ precursors using the “phase separation and evaporation” method. Resistive switching measurements of the prepared TiO_2_ nanostructures by conductive atomic force microscopy showed a stable ON/OFF ratio (~5 × 10^4^) at a reading voltage of −0.4 V (independent of the spatial position of the tip) (Figure 12b). Annealing the nanostructures at high vacuum temperatures promoted the loss and evaporation of Bi and greatly reduced the throughput rate by about 2 × 10^2^, since the annealing did not change other morphological and structural parameters, revealing the key role of Bi. Thus, it was concluded that these Bi ions acted as charge capture/release sites for carriers in the presence of an external electric field, which enhanced the resistive behavior (Figure 12c). This study suggests that the intentional introduction of defects or doping in oxide heteroepitaxy is an efficient way to achieve urgent electronic properties in metal oxide nanostructures.

## 3. Conclusions

The study of the literature published in recent years can greatly enhance our understanding of the properties and applications of TiO_2_ -based- nanomaterial-based optoelectronic devices, especially photodetectors, LEDs, memories, sensors, transistors, and other optoelectronic applications. Although TiO_2_ -based optoelectronic devices have shown outstanding performance over other materials, researchers are still testing TiO_2_ synthesis methods and combinations with different substrates to create more stable and exceptional optoelectronic performance. For example, different nanostructures of TiO_2_ are used as support materials for anchoring semiconductor particles to further improve the performance of optoelectronic devices. Researchers have combined TiO_2_ with conductive materials such as metals, metal oxides, carbonaceous materials, or doped TiO_2_ to enhance its performance. Low-cost dopants that promote electron transport and good electron mobility can enhance the photo-induced charge separation processes and magnetic properties in TiO_2_. Therefore, changing the electrode structure of the device, controlling the degree of doping of TiO_2_ based nanomaterials, and refining the details of the preparation process may be effective strategies for future optoelectronic applications.

There are few studies on TiO_2_based nanomaterial-based optoelectronic devices under extreme conditions. The stability and lifetime of optoelectronic devices under high-temperature and high-pressure conditions have been critical issues. In recent decades, although the various properties of electronic components have developed considerably, the life and functionality of components under specific temperature and pressure conditions will decrease as the temperature rises. In addition, temperature has a significant influence on the current transport mechanism, and many properties of the intrinsic carriers show temperature-dependent characteristics. Therefore, further research is needed to develop optical and electrical devices based on TiO_2_ under extreme conditions. Diamonds have a wide forbidden band (5.47 eV), high thermal conductivity, a low background current, physical and chemical stability at room temperature, etc., and are an excellent p-type conductive material for high-voltage, high-power, optoelectronic devices. Diamonds have a wide electrochemical window, which can provide a sufficient bias voltage to improve the photochemical performance of TiO_2_-based devices. Therefore, the fabrication of TiO_2_/diamond heterojunctions offers the possibility of improving the performance of photodetectors, light-emitting diodes, memories, sensors, transistors, and other instruments under extreme conditions.

## Figures and Tables

**Figure 1 nanomaterials-13-01141-f001:**
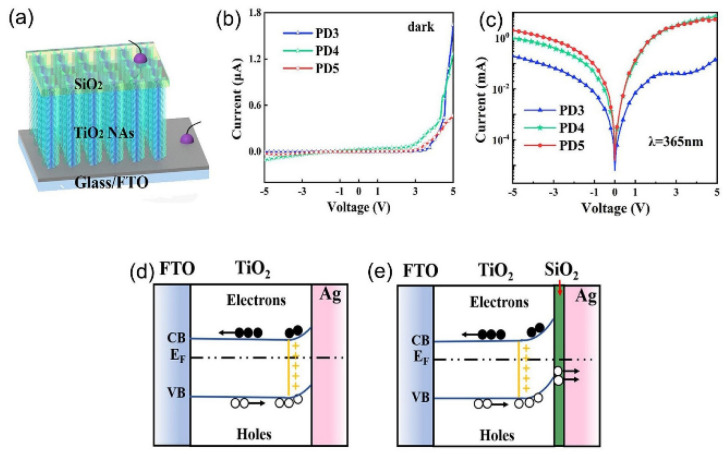
(**a**) The diagram of the TiO_2_/SiO_2_ mixing device. (**b**,**c**) *I*−*V* curves of TiO_2_/SiO_2_ hybrid devices under two conditions of darkness and light. (**d**,**e**) Energy band diagram of PD in TiO_2_ NAs and TiO_2_/SiO_2_ mixed devices. Reproduced with permission from Ref. [9]; published by Elsevier BV, 2021.

**Figure 2 nanomaterials-13-01141-f002:**
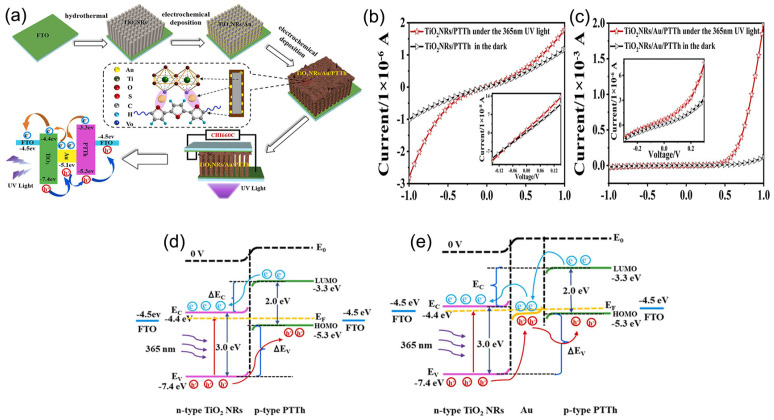
(**a**) Diagram of the preparation and assembly of a self-powered UV detector. Linear (**b**,**c**). (**d**,**e**). Self-powered mode of TiO_2_ NRs/PTTh and TiO_2_ NRs/Au/PTTh heterojunction energy band diagrams. Reproduced with permission from Ref. [11]; published by Elsevier BV, 2022.

**Figure 3 nanomaterials-13-01141-f003:**
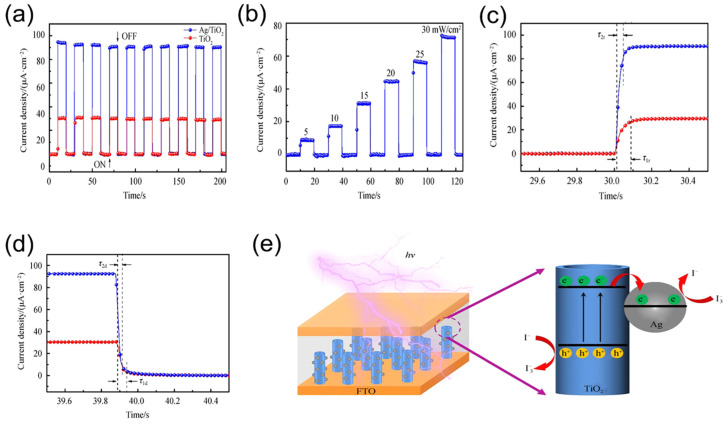
The patterns of time−dependent current response (**a**) measured under on/off UV illumination, current density (**b**) as a function of the incident UV light intensity, rising edges (**c**) and decaying edges (**d**), and (**e**) schematic diagram of the charge transfer of the UV detector based on Ag/TiO_2_ nanotubes. Reproduced with permission from Ref. [2]; published by Science Press, 2019.

**Figure 4 nanomaterials-13-01141-f004:**
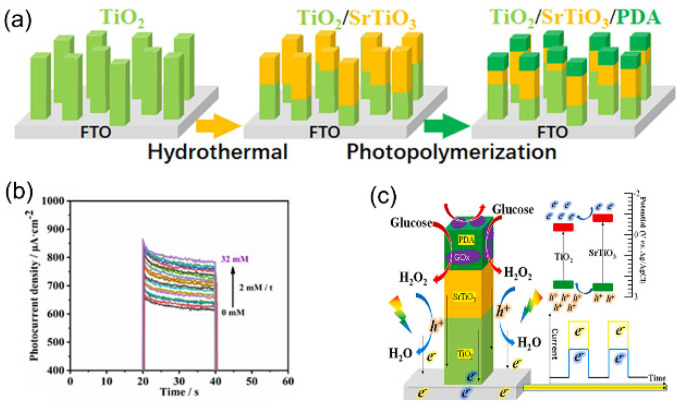
(**a**) Guidelines for the design of TiO_2_/SrTiO_3_/PDA electrodes. (**b**) TiO_2_/SrTiO_3_/PDA/GO_x_ in 0.1 M PBS (pH7.4) electrolyte to glucose concentration from 0-32 mM increasing photocurrent time density reaction. (**c**) Heterojunction mechanism of TiO_2_/SrTiO_3_ and reaction mechanism of TiO_2_/SrTiO_3_/PDA/GO_x_. Reproduced with permission from Ref. [36]; published by Elsevier Ltd., 2021.

**Figure 5 nanomaterials-13-01141-f005:**
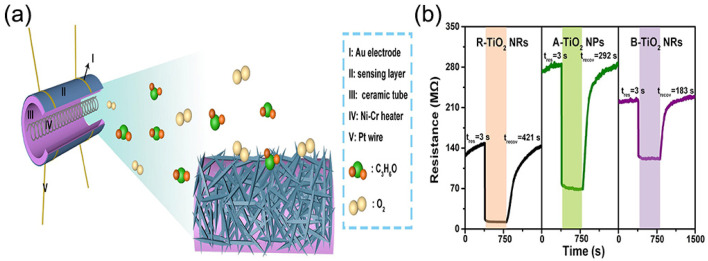
(**a**) Schematic diagram of sensor structure based on R-TiO_2_ NRs sample. (**b**) Single-cycle resistance transient curves of three nanoparticles to 100 ppm acetone at 320 °C. Reproduced with permission from Ref. [41]; published by Academic Press Inc., 2022.

**Figure 6 nanomaterials-13-01141-f006:**
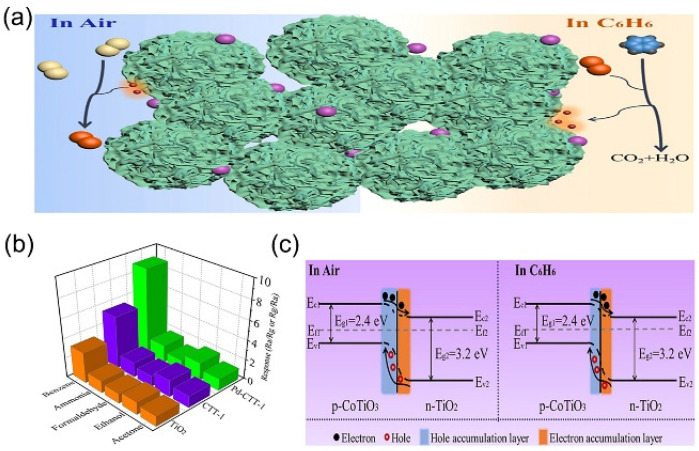
(**a**) Illustration of the sensor sensing mechanism. (**b**) Selectivity of PD-CTT-1 sensor for various test gases up to 10 ppm. (**c**) The energy band structure of CoTiO_3_ and TiO_2_. Reproduced with permission from Ref. [42]; published by Elsevier, 2022.

**Figure 7 nanomaterials-13-01141-f007:**
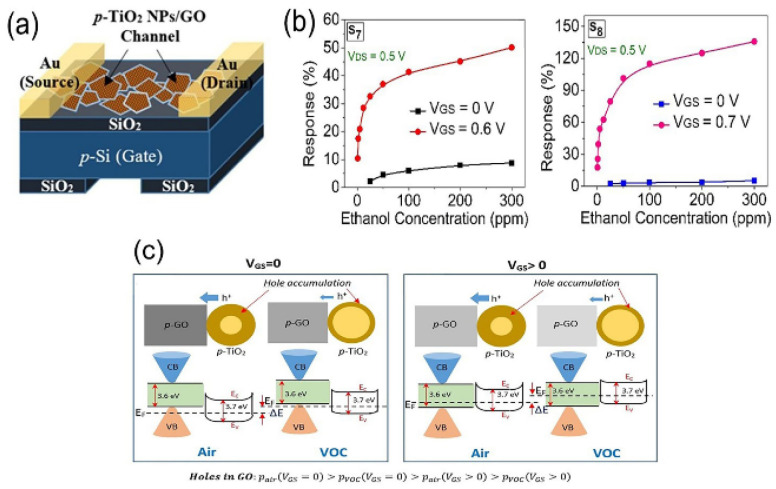
(**a**) Schematic diagram of a back gate FET embedded in the GO channel of p-TiO_2_ NPs. (**b**) Comparison of reaction amplitude between *V_GS_
*= 0 V and *V_GS_
*> 0 V in (g)S7 and (h)S8 is a function of ethanol concentration. (**c**) Schematic diagram of energy band in the case of *V_Gs_* = 0 and *V_Gs_* > 0. Reproduced with permission from Ref. [54]; published by Elsevier, 2021.

**Figure 8 nanomaterials-13-01141-f008:**
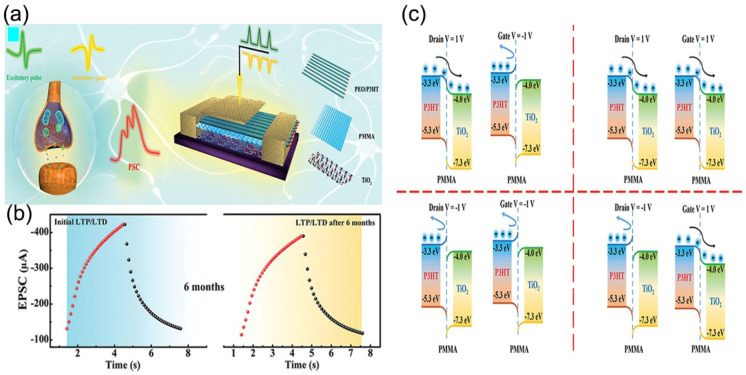
(**a**) Biological synapses and p-i-n JST schematic diagram. (**b**) The potential (30 pulses of 3 V) and inhibition (30 pulses of −4 V) of p-i-n JST at -1V *V_D_* before and after 6 months of exposure to natural environment. (**c**) Schematic diagram of ribbon bending of P3HT/TiO_2_ junction under bias voltage. Reproduced with permission from Ref. [57]; published by Wiley-Blackwell, 2021.

**Figure 9 nanomaterials-13-01141-f009:**
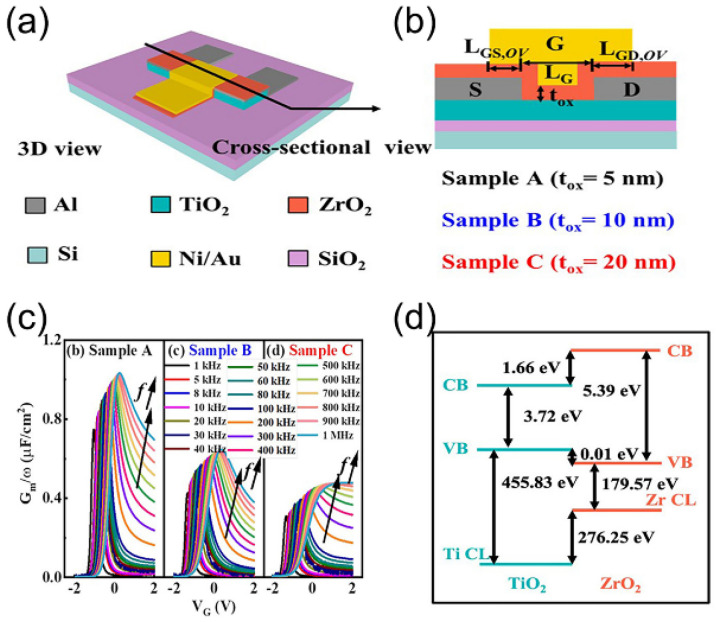
A TiO_2_ TFT with the same device size and different ZrO_2_ dielectric layer thickness in stereoscopic view (**a**) and cross-sectional view (**b**). (**c**) Equivalent parallel conductance (*G*/*ω*) measured for samples A, B and C as a function of applied voltage. (**d**) Derived band alignment of ZrO_2_/TiO_2_ from X−ray photoelectron spectroscopy (XPS) results. Reproduced with permission from Ref. [59]; published by the American Chemical Society, 2021.

**Figure 10 nanomaterials-13-01141-f010:**
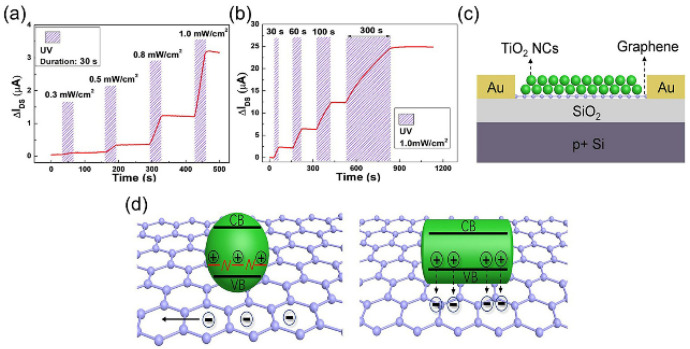
N-TiO_2_ device instantaneous photocurrent response to UV light (**a**) with different Ee values or (**b**) with different durations. (**c**) Schematic of a N-TiO_2_ device. (**d**) Schematic diagram of hole storage in N-TiO_2_ NC (green spheres) and schematic diagram of hole recombination in commercial TiO_2_ NC (green rods) with electrons in graphene after UV irradiation. Reproduced with permission from Ref. [75]; published by the American Chemical Society, 2020.

**Figure 11 nanomaterials-13-01141-f011:**
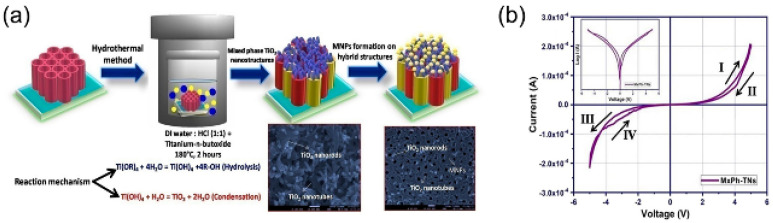
(**a**) MxPh−TNs and MX−TNHs coated with MNPs by the photo deposition process. (**b**) *I−V* hysteresis return line of Al/MxPh−TNs/Ti. Reproduced with permission from Ref. [77]; published by the American Chemical Society, 2020.

**Figure 12 nanomaterials-13-01141-f012:**
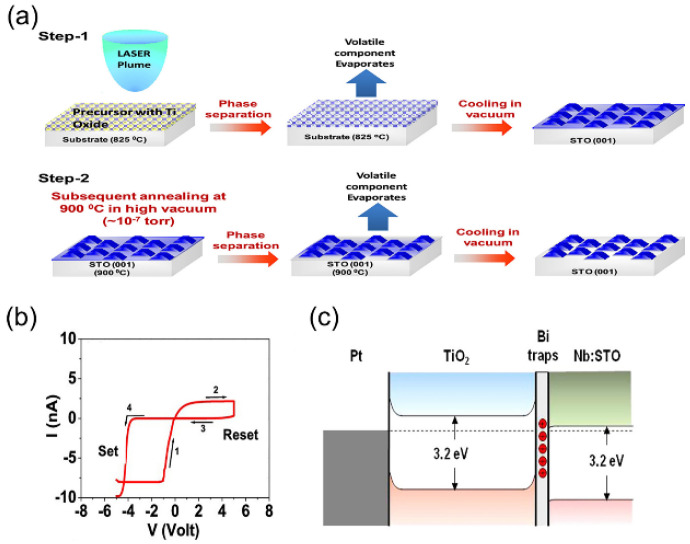
(**a**) Schematic diagram of titanium dioxide nanoparticle synthesis technology. (**b**) Pt/TiO_2_/Nb volt ampere characteristics of SrTiO_3_ (STO) devices. (**c**) The energy band diagram of a Pt/TiO_2_/Nb:STO device during thermal equilibrium. Reproduced with permission from Ref. [79]; published by the American Chemical Society, 2020.

## Data Availability

The data presented in this study are contained within the article.

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
