# Peer review of "A Review on the Progress of Optoelectronic Devices Based on TiO2 Thin Films and Nanomaterials"

_nanomaterials, 2023, doi:10.3390/nano13071141_

Round 1

Reviewer 1 Report

Shunhao Ge et al., reports about the recent progress of TiO2 nanomaterials for optoelectronic applications. Although the authors did collected thorough literature survey, the reviewer find that the overall manuscript should be revised further to clearly deliver the idea of the manuscript to readers. I suggest the following revision.

1. The writing should be revised. There are multiple expressions in which the meaning is not clear. For example, the sub-section title  "2.1.1 UV photodetectors for nanoparticle structures" , or the expression TiO2/SiO2 mixing device (at line 102) doesn't make sense.

2. The abstract mentions about photocatalyst or electrochemical energy applications, which are different topic than the optoelectronic devices. The manuscript should be focused to the single topic, even though TiO2 is a versatile material and widely used in various fields.

3. Some figures are too small and not readable, such as Figure 2, 3, 5, or 8.

4. I don't understand the meaning of "Junction type field effect transistors" at line 419.

Overall, I recommend to revise the manuscript so that all the ideas relate to a single topic - TiO2 for optoelectronic device. It may be OK to discuss about the other advantages of TiO2 such as energy applications, it should not distract the readers from the main topic. Additionally, many paragraphs are too long to read and get an idea. English editing from professional agent is recommended for better clarity.

Reviewer 2 Report

In this review, Ge et al. propose a state of the art review on TiO2-based optoelectronic applications.

As the authors write, it is true that a review on these aspects is lacking. However, the objective of the authors to underline the difficulties and opportunities that are attached to this thematic does not seem to be reached.

The article is divided in several sections (UV photodetectors for nanoparticle structures..) and for each section 2-3 articles are summarized without really understanding why these articles were chosen and their significances compared to the others. As a result, each section seems to be a summary of abstracts and the review lacks of overall scientific vision that guides the reader.

The section titles do not always seem appropriate: shouldn't UV detectors for nanoparticles structures be named UV detectors based on nanoparticles?

Some articles do not seem to be in the right section (monolayer graphene photodetector on the section for 1D? and many other examples).

It is also problematic that many references are not numbered: all the references numbers for each reference which is introduced by Name et al. is missing on the same sentence. The reference “Raduban et al” (p 4) does not seem to be in the bibliography.

Some acronyms do not seem to be explained (MZO PD?) and a list of acronyms at the end of the articles would be welcome.

The figures are generally small and not very readable.

In general, the review does not really convince of its usefulness for the community.

Round 2

Reviewer 2 Report

A lot of corrections were made, in particular of english, and the ms can be published now.